# A putative ATPase mediates RNA transcription and capping in a dsRNA virus

Xuekui Yu[1,2]*, Jiansen Jiang[1,2], Jingchen Sun[1,3]*, Z Hong Zhou[1,2]*

[1]Department of Microbiology, Immunology and Molecular Genetics, University of California, Los Angeles, Los Angeles, United States; [2]California Nanosystems Institute, University of California, Los Angeles, Los Angeles, United States; [3]Subtropical Sericulture and Mulberry Resources Protection and Safety Engineering Research Center, Guangdong Provincial Key Laboratory of Agro-animal Genomics and Molecular Breeding, College of Animal Science, South China Agricultural University, Guangzhou, China

**Abstract** mRNA transcription in dsRNA viruses is a highly regulated process but the mechanism of this regulation is not known. Here, by nucleoside triphosphatase (NTPase) assay and comparisons of six high-resolution (2.9–3.1 Å) cryo-electron microscopy structures of cytoplasmic polyhedrosis virus with bound ligands, we show that the large sub-domain of the guanylyltransferase (GTase) domain of the turret protein (TP) also has an ATP-binding site and is likely an ATPase. S-adenosyl-L-methionine (SAM) acts as a signal and binds the methylase-2 domain of TP to induce conformational change of the viral capsid, which in turn activates the putative ATPase. ATP binding/hydrolysis leads to an enlarged capsid for efficient mRNA synthesis, an open GTase domain for His217-mediated guanylyl transfer, and an open methylase-1 domain for SAM binding and methyl transfer. Taken together, our data support a role of the putative ATPase in mediating the activation of mRNA transcription and capping within the confines of the virus.

*For correspondence: xuekuiyu@ucla.edu (XY); cyfz@scau.edu.cn (JS); Hong.Zhou@UCLA.edu (ZHZ)

Competing interests: The authors declare that no competing interests exist.

## Introduction

Viral transcription is highly regulated, as demonstrated biochemically in viruses of the *Reoviridae* (*Shatkin and Sipe, 1968*; *Furuichi, 1974*, *1978*; *Borsa et al., 1981*; *Farsetta et al., 2000*). mRNA transcription in these viruses is activated by external actions, for example, removal of their outer shell in multi-shelled reoviruses (*Shatkin and Sipe, 1968*; *Borsa et al., 1981*; *Farsetta et al., 2000*) and binding of S-adenosyl-L-methionine (SAM) in the single-shelled cytoplasmic polyhedrosis virus (CPV) (*Furuichi, 1974*, *1978*). The outer shell and the binding sites of SAM are far away from the RNA-dependent RNA polymerases (RdRPs) inside the virus. How these external actions regulate viral mRNA transcription has been a mystery.

Viruses in the *Reoviridae* contain 9–12 segments of dsRNA enclosed within an inner core that is a self-competent molecular machine fully capable of RNA transcription and processing (*Mertens et al., 2004*; *Zhou, 2008*). Each of the 9–12 dsRNA segments wraps around an RdRP located underneath an icosahedral vertex and can undergo independent and simultaneous RNA transcription within an intact core (i.e., endogenous RNA transcription) (*Smith and Furuichi, 1982*). The simplest of these, the single-shelled CPV (*Zhou, 2008*) has been used as a model system for viral RNA transcription and high-resolution cryo-electron microscopy (cryoEM) studies, as highlighted by the discovery of mRNA cap structures (*Furuichi, 1974*; *Furuichi and Miura, 1975*) and the demonstration of near atomic resolution cryoEM (*Yu et al., 2008*).

To find out how viral mRNA transcription is regulated, we set out to determine a series of structures of CPV in complex with different ligands at resolutions ranging from 2.9 to 3.1 Å. We

**eLife digest** Viruses can only replicate by invading the cells of other organisms, such as plants and animals. Each virus carries genetic material in the form of molecules of DNA or ribonucleic acid (RNA), which are packaged in a shell made of proteins.

The cytoplasmic polyhedrosis virus has a genome made of a type of RNA called double-stranded RNA. Once inside a host cell, sections of the virus genome are copied to make molecules of 'messenger RNA' in a process called transcription. Small chemical groups called guanylyl and methyl groups are added to the messenger RNAs before they are used as templates to make the virus proteins.

A small molecule called S-adenosyl-L-methionine (SAM) can activate transcription of the virus genome by binding to a protein called turret in the shell of the virus. The turret protein is involved in adding the guanylyl and methyl groups to the messenger RNA molecules, but it is not clear how the protein activates transcription.

Here, Yu et al. used a technique called cryo electron microscopy to study how the virus binds SAM to activate transcription. The experiments show that the binding of SAM to one region or 'domain' of the turret protein leads to changes in the virus shell. This enables another domain of the turret protein to bind a small molecule called ATP and break it down. The energy released from breaking down ATP causes further changes of the shell of the virus to activate transcription and the addition of guanylyl and methyl groups to the newly made messenger RNAs.

In the future, experiments that directly observe the RNA inside each virus shall offer fresh insights as to how the genomes of cytoplasmic polyhedrosis virus and other similar viruses are transcribed.

discovered that the large sub-domain of guanylyltransferase (GTase) domain of CPV turret protein (TP) also has an ATP-binding site and is likely an ATPase that mediates the activation process of viral RNA transcription and capping. This process involves sensing the presence of the signal molecule SAM by methylase −2 (MT-2) domain of CPV TP, activating the putative viral ATPase, enlarging the viral capsid for efficient mRNA syntheses, and opening the GTase and MT-1 to enable guanylyl and methyl transfer.

## Results

### CryoEM structures at up to 2.9 Å and visualization of ligands

To reveal the mechanisms of transcriptional regulation of viruses within the *Reoviridae* family, we determined the cryoEM structures of six CPV/ligand complexes in the presence of magnesium ion: CPV+SAM (i.e., 'S-CPV'), CPV+SAM+4 nucleoside triphosphates (NTPs) (i.e., transcribing, or 't-CPV'), CPV+SAM+GTP+ATP (i.e., 'SGA-CPV'), CPV+SAM +GTP (i.e., 'SG-CPV'), CPV+GTP (i.e., 'G-CPV'), and CPV+ATP (i.e., 'A-CPV') at resolutions ranging from 2.9 to 3.1 Å (*Figure 1A–C*, *Table 1*, *Video 1* and *Figure 1—figure supplement 1*). This range of resolutions has permitted us to identify side chains of amino acid residues and to define conformations of bound ligands to build atomic models. Like the atomic model of unliganded CPV (*Yu et al., 2011*), the atomic models of these liganded CPV all contain two conformers of the capsid shell proteins (CSP-A and CSP-B), two conformers of the large protrusion proteins (LPP-3 and LPP-5), and one conformer of TP in each asymmetrical unit (e.g., *Figure 1B*). We show below that the large sub-domain of GTase domain of TP (*Zhou et al., 2003*; *Yu et al., 2008*) also has an additional ATP-binding site and is likely an ATPase (*Figure 1D,E*). Except for A-CPV, these atomic models also contain ligands revealed in our cryoEM maps (*Table 1*). In S-CPV, one SAM binds to the MT-2 domain of each TP. In G-CPV, one GTP binds to the GTase site of GTase domain. In SG-CPV, two SAM molecules bind to the MT-1 and MT-2 domains, one $Mg^{2+}$-GTP to the GTase site and one GTP to the putative ATPase site. In t-CPV and SGA-CPV, two SAM molecules bind to the MT-1 and MT-2 domains (*Zhu et al., 2014*), one $Mg^{2+}$-GTP to the GTase site and one ATP to the putative ATPase site. As we will report in detail below, a comparison of these structures and their correlation with the accompanying biochemical results have led to our discovery of a putative ATPase-mediated regulation process for activating viral RNA transcription and capping.

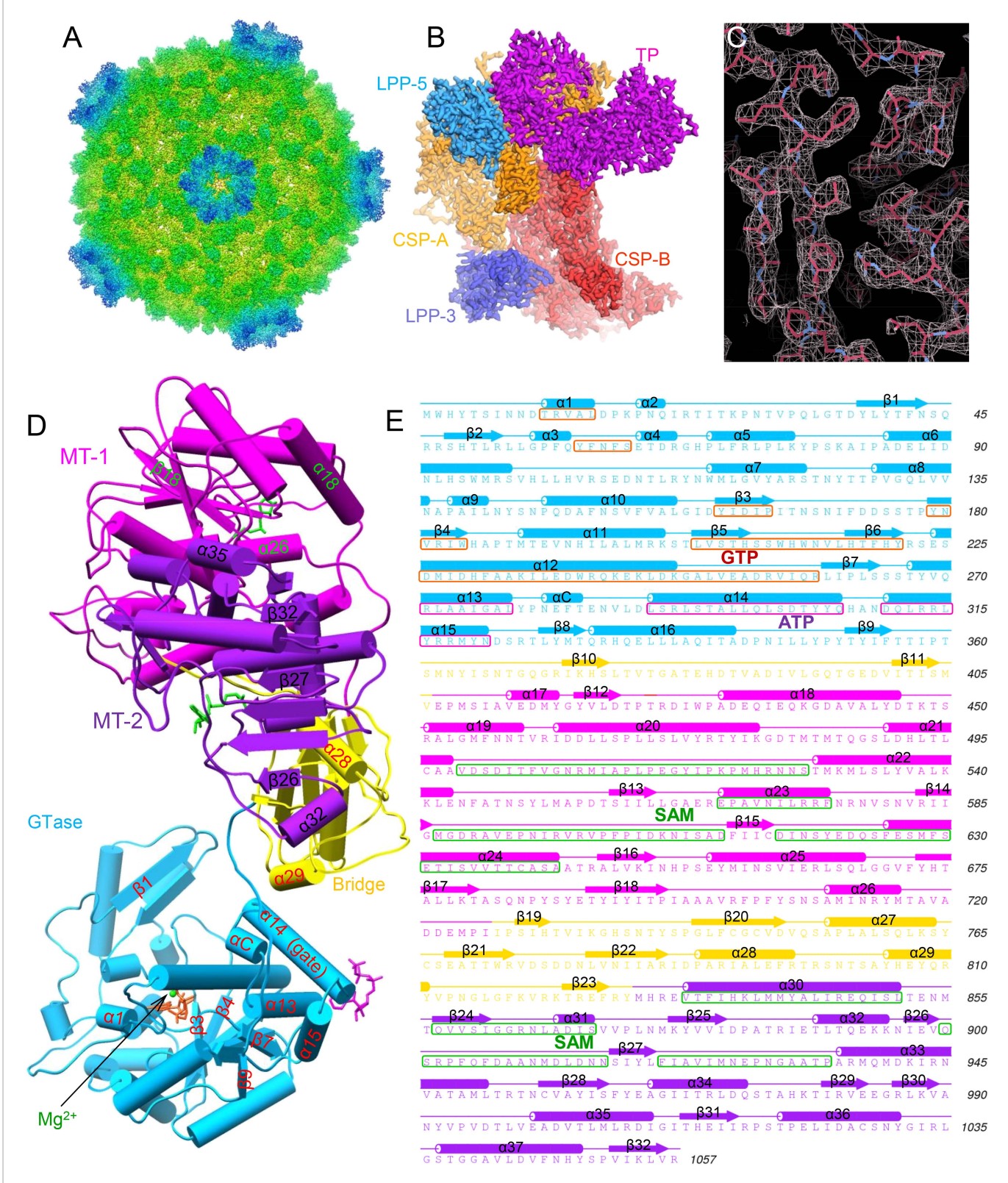

**Figure 1**. Structural overviews of cytoplasmic polyhedrosis virus (CPV) bound with different ligands involved in regulation and capping for viral RNA transcription. (**A**) Radially colored G-CPV reconstruction at 2.9 Å resolution as viewed along a fivefold axis. (**B**) Density map of an asymmetric unit of G-CPV is colored by protein subunit. (**C**) Density map (mesh) and atomic model (stick) of a selected region from CSP-A of G-CPV, showing characteristic side

*Figure 1. continued on next page*

*Figure 1. Continued*

chains. (**D**) Structures of turret protein (TP) and ligands in t-CPV. TP is colored by domain. The $Mg^{2+}$ and GTP in the guanylyltransferase (GTase) site are in green and orange, respectively; ATP in the putative ATPase site is in magenta; the two S-adenosyl-L-methionines (SAMs) in MT-1 and MT-2 are in green. (**E**) Schematic illustration of t-CPV TP structure. Secondary elements involved in hydrogen bonding or stacking interactions with GTP and ATP are highlighted in orange red and magenta, respectively. Secondary elements involved in interactions with SAM are highlighted in green.

The following figure supplement is available for figure 1:

**Figure supplement 1**. Resolution assessment of CPV particle reconstructions.

## SAM alone triggers slight global protein movements

Because SAM is required for efficient mRNA synthesis in CPV in addition to being the methyl donor for mRNA methylation (*Furuichi, 1974*, *1978*), we first asked whether the presence of SAM would have any effect on the structure of CPV. Superposition of the structures (both at 3.1 Å resolution) of S-CPV and the unliganded CPV (*Yu et al., 2011*) shows that the capsid shell of S-CPV is slightly expanded with a non-uniform outwards movement of all structure proteins (*Figure 2A,B*, *Video 2* and *Figure 2—figure supplement 1A–C*). (In contrast, the A-CPV structure reported here does not have such capsid expansion and protein movements, see below.) For example, the apical domain of CSP-A, located next to the fivefold axis, has the largest movement of ~1 Å (RMSD: 0.97 Å); the dimerization domain, located near the twofold axis, has the smallest movement of ~0.5 Å (RMSD: 0.49 Å); the CPV-unique small protrusion domain, which is located between the apical and the dimerization domain, moves outwards ~0.8 Å (RMSD: 0.83 Å) (*Figure 2A* and *Video 3*). TP, residing on the apical domain of CSP-A, moves outwards ~1 Å (RMSD: 0.99 Å), which is the same as the displacement of the apical domain of CSP-A (*Figure 2B*). In CSP-B, located around the threefold axis, the outwards movements of the apical, small protrusion, and dimerization domains are ~0.85 Å (RMSD: 0.85 Å), 0.6 Å (RMSD: 0.57 Å), and 0.5 Å (RMSD: 0.49 Å), respectively (*Figure 2—figure supplement 1A*). Accordingly, the movement (RMSD: 0.67 Å) of LPP-5 is slightly larger than that of LPP-3 (RMSD: 0.43 Å) (*Figure 2—figure supplement 1B,C*).

Both MT-1 and MT-2 domains of TP have the typical structural motif of SAM-dependent methyltransferases with a seven-stranded β-sheet sandwiched by α-helices (*Schluckebier et al., 1995*; *Hodel et al., 1996*; *Reinisch et al., 2000*; *Sutton et al., 2007*). Unexpectedly, only the MT-2 domain

**Table 1**. CryoEM imaging and model refinement statistics

| Sample name | S-CPV | t-CPV | SGA-CPV | SG-CPV | G-CPV | A-CPV |
|---|---|---|---|---|---|---|
| **CryoEM reconstruction** | | | | | | |
| Particles included in the final reconstruction | 44,908 | 41,624 | 40,898 | 46,147 | 71,946 | 19,447 |
| Resolution (Å) | 3.1 | 3 | 3.1 | 3.1 | 2.9 | 3.1 |
| Bound ligands | One SAM bound to MT-2 | SAMs bound to MT-1 and MT-2; one Mg-GTP and one ATP bound to GTase domain | Identical to those of t-CPV | SAMs bound to MT-1 and MT-2; one Mg-GTP bound to GTase site; one GTP to ATPase site | One GTP bound to the GTase site of GTase domain | No ATP bound |
| Structural changes | Structure protein movements outwards | Structure protein movements outwards and local conformational changes | Identical to those of t-CPV | Identical local conformational changes; different global protein movements | No changes | No changes |
| Model refinement | | | | | | |
| Resolution range (Å) | 40–3.1 | 40–3.0 | 40–3.1 | 40–3.1 | 40–2.9 | 40–3.1 |
| R-factor (%) | 19.85 | 19.74 | 19.78 | 18.25 | 19.93 | 19.51 |

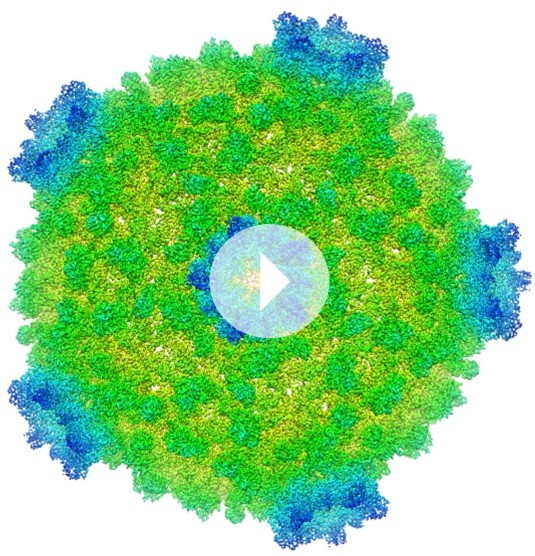

**Video 1.** Radially colored G-CPV reconstruction at 2.9 Å resolution as viewed along a fivefold axis.

in S-CPV bound SAM (*Figure 2C,D* and *Figure 2—figure supplement 1D*). Except for slight displacement due to the outwards movement of TP as described above, the MT-2 domain structure of S-CPV is indistinguishable from that of the unliganded CPV at the current resolution of 3.1 Å (*Figure 2E*).

## Structure changes in t-CPV and discovery of an ATP-binding site in TP

The above observed SAM-triggered conformational change correlates with previous biochemical data establishing a role of SAM in inducing mRNA synthesis (*Furuichi, 1974, 1978*). In order to find out how SAM does this, we obtained a structure of t-CPV at 3.0 Å resolution, that is, virions incubated with SAM, 4 NTPs, and Mg$^{2+}$. In contrast to that of S-CPV (*Figure 3A*), the cryoEM image of t-CPV shows string-like densities emanating from the viral particles, which we attribute to newly synthesized mRNA molecules in the process of release from the actively transcribing virions (arrows in *Figure 3B*). However, no mRNA densities are visible in our icosahedral reconstruction because these RNA molecules are transcripts of different genomic segments at different stages of the dynamic transcription process and are smeared by averaging.

Structural comparison between t-CPV and S-CPV reveals conformational changes of the capsid proteins in t-CPV (*Figure 3C,D*, *Video 4* and *Figure 3—figure supplement 1A–C*). Among the five protein molecules within each asymmetric unit of CPV, only LPP-3 remains unchanged in both the location and structure (*Figure 3—figure supplement 1A*) and the other four protein molecules exhibited changes in their locations, their structures, or both. The locations of LPP-5 molecules in t-CPV and S-CPV differ although their structures are the same (*Figure 3—figure supplement 1B*), indicating a rigid-body type of movement, likely effected by changes of the underlying CSP molecules. By contrast, the other molecules undergo both global domain movements and local conformation changes from S-CPV to t-CPV (*Figure 3C,D* and *Figure 3—figure supplement 1C*). All domains of TP undergo global outwards movements (~9 Å), while only the MT-1 domain and the large sub-domain of GTase domain of TP exhibit local conformation changes (*Figure 3C*). CSP-A not only rotates outwards (up to 9 Å) around a pivot point near the twofold axis (global domain movements indicated by dashed ellipses in *Figure 3D* and *Video 5*), but it also changes conformation in its apical domain (dotted ellipse in *Figure 3D*). CSP-B also undergoes similar but less obvious changes than CSP-A (*Figure 3—figure supplement 1C*). These structural changes in CSP molecules result in an enlarged, yet stable capsid of the transcribing CPV. Since viral mRNA synthesis takes place within the confines of intact virus core, an enlarged capsid would facilitate dsRNA template movement, enabling efficient mRNA synthesis.

The t-CPV structure contained two ligands bound to the GTase domain of each TP (*Figure 1D*, *Video 6* and *Figure 3—figure supplement 1D*). The first is the expected GTP molecule involved in transfer of a guanylyl group catalyzed by GTase and is located at a cleft of the GTase active site (*Video 6* and *Figure 3—figure supplement 1D*). The second is unexpected ligand, bound to the large sub-domain, away from the cleft (*Video 6* and *Figure 3—figure supplement 1D*).

The density of the unexpected ligand is as strong as that of the GTP bound to the GTase site and the surrounding amino acid residues, and it fits very well with the atomic model of an ATP molecule, suggesting that the large sub-domain of GTase domain of TP could also be a viral ATP-binding site (*Figure 4A* and *Video 7*). Located outside the turret chamber (*Figures 1D, 4A* and *Figure 4—figure supplement 1*), the putative ATP-binding site is inaccessible to both the dsRNA genome and the

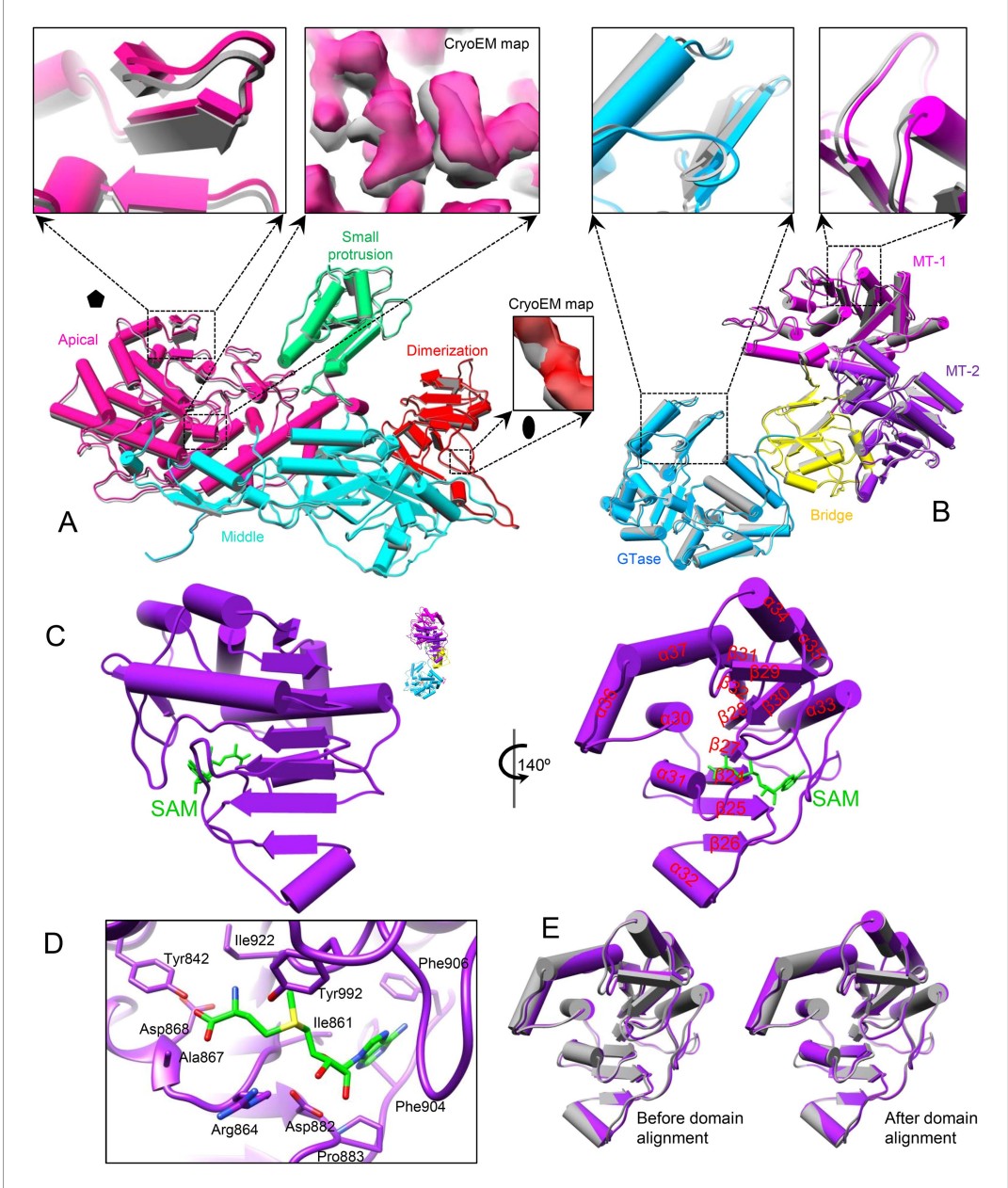

**Figure 2**. SAM alone binds to MT-2 of TP and triggers global movement of all capsid proteins. (**A**) Superimposition of CSP-A between unliganded CPV (gray) and S-CPV (colored by domain). Insets: zoom-in views of the boxed regions. The twofold and fivefold axes are indicated by a pentagon and an oval, respectively. (**B**) Superimposition of TP between unliganded CPV (gray) and S-CPV (colored by domain as in *Figure 1D*). Insets: zoom-in views of the boxed regions from GTase and MT-1 domains, respectively. (**C**) Structure of MT-2 (purple) and SAM (green). Left, view as the guide map (inset). Right, view rotated as indicated. (**D**) Active site of MT-2. SAM is colored by element: carbon in green, nitrogen in blue, oxygen in red, and sulfur in yellow. Side chains of those amino acids interacting with SAM are shown. (**E**) Superimposition of MT-2 between unliganded CPV (gray) and S-CPV (purple) before (left) and after (right) domain alignment using Cα positions.

The following figure supplement is available for figure 2:

**Figure supplement 1**. Global movement of viral capsid proteins caused by SAM bound to the externally located MT-2.

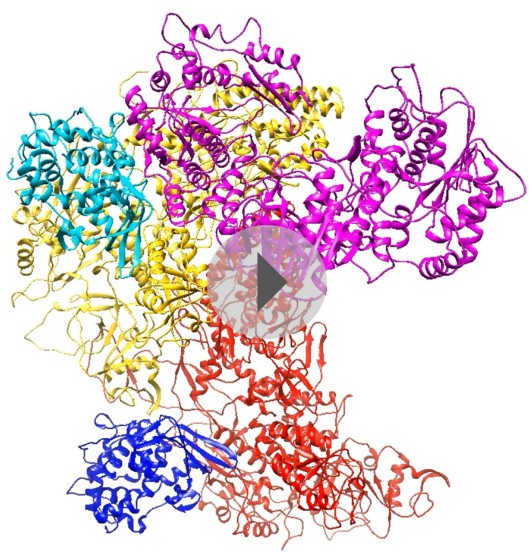

**Video 2.** Conformational changes from unliganded CPV to S-CPV. Atomic model of an asymmetric unit is colored by protein subunit as in *Figure 1B*.

nascent mRNA, thus rendering it unable to directly participate in the mRNA transcription and the capping reactions but may function as a regulatory protein or enzyme.

To establish the identity of the aforementioned ligand and the function of the putative ATP-binding site, we subsequently performed the following two structural studies.

First, through incubating CPV capsids with SAM, GTP, ATP, and $Mg^{2+}$, we obtained the SGA-CPV particle which, lacking of UTP and CTP, is incapable of mRNA transcription. Indeed, under cryoEM, SGA-CPV (*Figure 4—figure supplement 2A*), and S-CPV (*Figure 3A*) particles look similar and differ from actively transcribing t-CPV particles (*Figure 3B*). However, the 3.1 Å structure of SGA-CPV shows global movements and local conformational changes of its structure proteins that are indistinguishable from those of t-CPV (*Figure 4A,B* and *Figure 4—figure supplement 2B*). Furthermore, the large sub-domain of SGA-CPV GTase domain also contains a ligand similar to that in the t-CPV (*Figure 4A,B* and *Figure 4—figure supplement 2C*).

Second, to eliminate the possibility of GTP as the ligand bound to the putative ATP-binding site in t-CPV, we obtained a 3D reconstruction of SG-CPV at 3.1 Å resolution. While the local conformational changes of SG-CPV structure proteins are identical to those of t-CPV (*Figure 4C*), the global movements of structure proteins of SG-CPV are slightly less than those in t-CPV or SGA-CPV (*Figure 4—figure supplement 3A*). For example, the movement of GTase domain in SG-CPV is ~1 Å less than that of t-CPV or SGA-CPV (*Figure 4D* and *Video 8*). Most importantly, the density of the bound GTP is not as strong as that of the ligand in t-CPV or SGA-CPV, and its triphosphate group becomes invisible when displayed at the same threshold of 3σ (*Figure 4C* and *Figure 4—figure supplement 3B*).

These results indicate that (1) the global movements and local conformational changes of structural proteins observed in t-CPV are not a consequence, but rather a trigger of RNA transcription; (2) the large sub-domain of the GTase domain binds ATP to mediate the conformational changes observed in t-CPV. Consistent with this assignment, only the large sub-domain (the one containing the ATP-binding site) of the GTase domain undergoes significant conformational changes between S-CPV and t-CPV (*Figures 3C, 4E*). Accompanying these conformational changes, part of the loop connecting α13 and α14 in S-CPV became a helix (αC) in SGA-CPV, SG-CPV, and t-CPV (*Figure 4*).

## Demonstration of viral ATPase activity and its SAM-dependence

Previous biochemical studies have shown that the hydrolysis of ATP is required for mRNA synthesis (*Furuichi, 1978*) and that efficient synthesis of CPV mRNA depended on the concentrations of SAM and ATP in a synergistic manner (*Furuichi, 1981*). We reason that the large sub-domain with the ATP-binding site is possibly an ATPase, and the synergy between SAM and ATP reflects

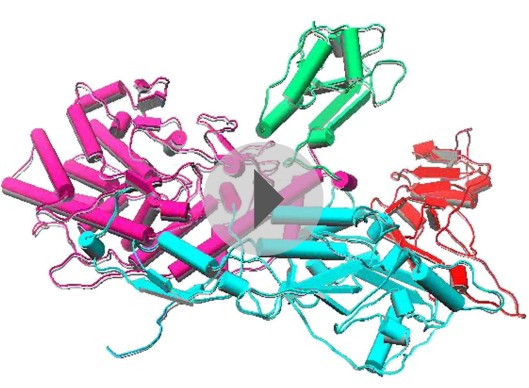

**Video 3.** Global movements of CSP-A caused by SAM bound to the externally located MT-2. Superimposition of CSP-A between unliganded CPV and S-CPV. CSP-A from unliganded CPV is in gray. CSP-A from S-CPV is colored by domain as in *Figure 2A*.

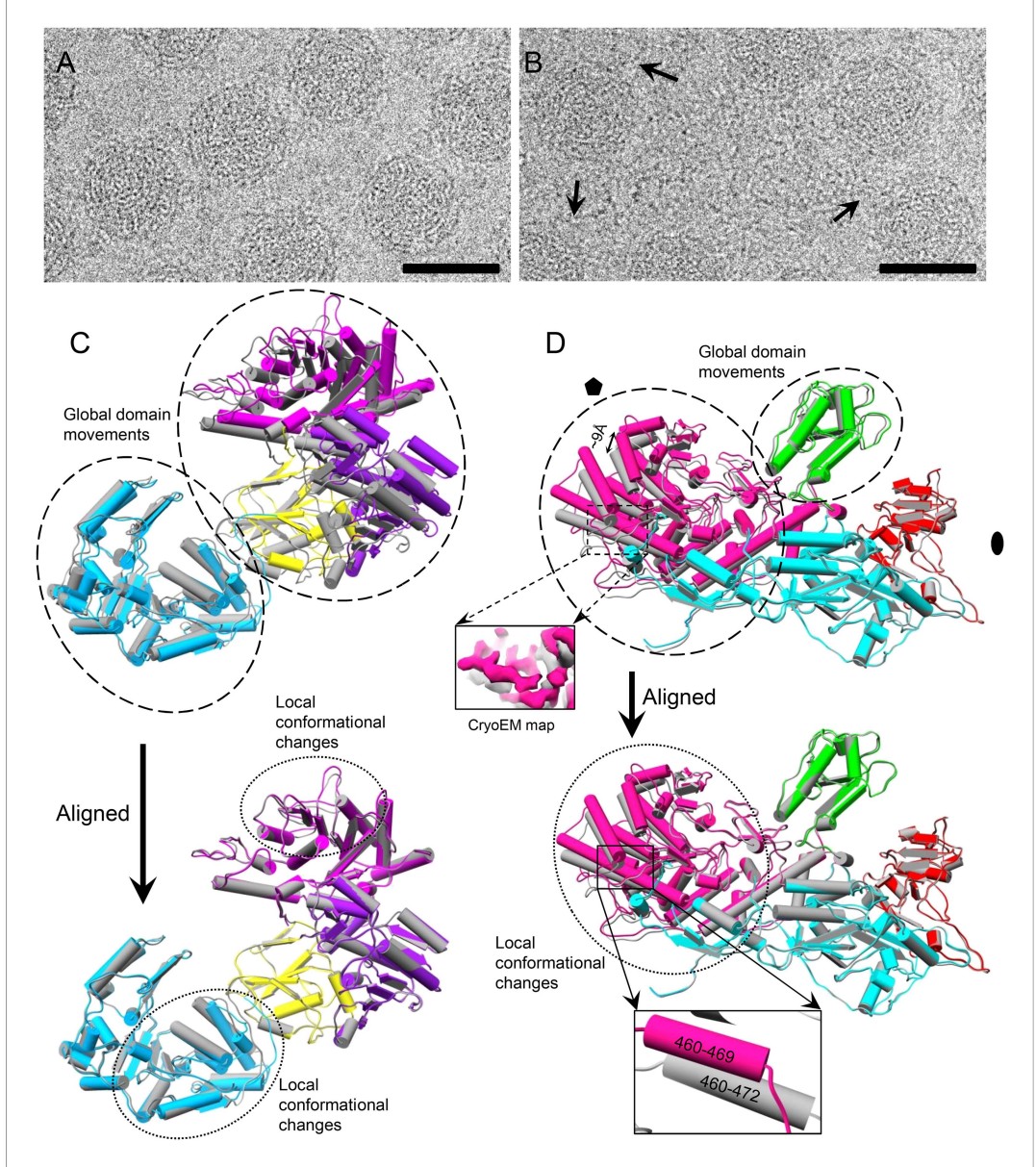

**Figure 3**. Comparison of S-CPV and t-CPV reveals global protein movements and local conformational changes. (**A**, **B**) Cryo-electron microscopy (cryoEM) images of S-CPV and t-CPV. Unlike that of S-CPV (**A**), the cryoEM image of t-CPV (**B**) shows characteristic string-like densities emanating from virus particles (arrows). Scale bars, 50 nm. (**C**) Superimposition of TP between S-CPV (gray) and t-CPV (colored by domain as in *Figure 1D*). Upper, domains that show global movements are indicated by dashed ellipses. Lower, GTase domain of t-CPV was aligned to that of S-CPV using Cα positions for residues in small sub-domain. Each of other three domains in t-CPV was aligned to its counterpart in S-CPV using Cα positions for residues in each domain. Regions that undergo local conformational changes are indicated by dotted ellipses. (**D**) Superimposition of CSP-A between S-CPV (gray) and t-CPV (colored as in *Figure 2A*). Upper, domains that show global movements are indicated by dashed ellipses. Inset, density maps of S-CPV (gray) and t-CPV (pink) from the boxed region. Lower, molecules were aligned using Ca positions for residues in small protrusion, middle and dimerization domains. Region that undergoes local conformational change is indicated by dotted ellipse. Part (470–472) of a helix (residues 460–472) in S-CPV becomes a loop in t-CPV (inset).

The following figure supplement is available for figure 3:

**Figure supplement 1**. Global movements and local conformational changes of capsid proteins observed in t-CPV. DOI: 10.7554/eLife.07901.012

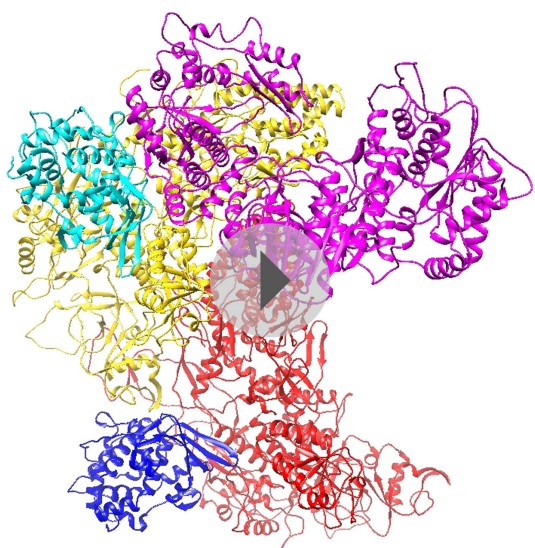

**Video 4.** Conformational changes from S-CPV to t-CPV. Atomic model of an asymmetric unit is colored by protein subunit as in *Figure 1B*.

a dependence of its activity on the presence of SAM. To test this hypothesis, we first obtained the 3D reconstructions of G-CPV and A-CPV at 2.9 Å and 3.1 Å resolutions, respectively (*Figure 1A–C*, *Table 1*, *Video 1* and *Figure 1—figure supplement 1*). Our structures of G-CPV and A-CPV show that, in the absence of SAM, neither GTP nor ATP induced any conformational change (*Table 1*). While a GTP bound to the GTase active site in G-CPV (*Figure 5A* and *Figure 5—figure supplement 1*), neither ATP nor GTP was observed at the newly discovered ATP-binding site in A-CPV and G-CPV (*Table 1*), indicating that ATP/GTP binding to the large sub-domain of TP GTase domain is directly regulated by SAM, most likely via binding to the MT-2 domain, since the structure of S-CPV revealed only MT-2 domain contained SAM (*Figure 2C,D*).

We then determined the rates of NTP hydrolysis of CPV in the presence and absence of SAM. Phosphate released upon hydrolysis of NTPs indeed depended on the presence of SAM and the most favorable NTP substrate was ATP, with decreasing rate of hydrolysis of other substrates in the order of GTP > CTP > UTP in the presence of SAM (*Figure 5B*). Previous biochemical studies have shown that the mRNA transcription of CPV is SAM-dependent and is specifically coupled to ATP hydrolysis (*Furuichi, 1978*, *1981*). Additionally, it has been shown that the removal of turret in orthoreovirus leads loss of mRNA transcription activity (*Luongo et al., 2002*). Our structural results, when combined with these biochemical data, suggest that the ATP-binding site of TP GTase domain is possibly a SAM-dependent ATPase that mediates the activation of mRNA transcription.

## The putative viral ATPase regulates the activities of GTase and MT-1

The density for the bound GTP in the GTase site is strong in t-CPV (*Figure 6A*, *Video 6* and *Figure 6—figure supplement 1A*) but weak in G-CPV, particularly at the triphosphate group (*Figure 5A* and *Figure 5—figure supplement 1*).

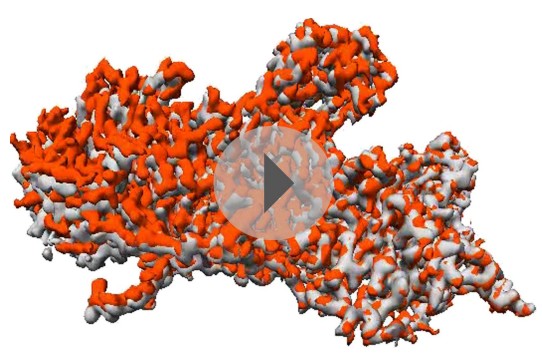

**Video 5.** Global movements and local conformational changes of CSP-A from S-CPV to t-CPV. Density map superimposition of CSP-A between S-CPV and t-CPV. CSP-A of S-CPV is in gray. CSP-A of t-CPV is in orange red.

Compared to that of G-CPV, the opening (or 'gate') leading to the GTase site and coupled to the putative mRNA releasing hole (*Yu et al., 2011*) is widened from 13 to 15 Å in t-CPV, likely to accommodate nascent mRNA (*Figures 5A, 6A,B*). Following the nomenclature of PBCV-1 GTase (*Hakansson et al., 1997*), we designate the states of GTase domain in G-CPV and t-CPV as closed and open states, respectively. In the open state of GTase domain in t-CPV, we observed extra density next to the β, γ phosphates of the bound GTP (*Figure 6A* and *Video 6*), which we interpret as coordinated $Mg^{2+}$ for two reasons. First, $Mg^{2+}$ was the only divalent cation in our reaction mixture. Second, the GTase domains in SGA-CPV and SG-CPV are also in the open state with prominent densities attributable to $Mg^{2+}$ (*Figure 6C,D* and *Video 9*). Biochemical data have shown that the $Mg^{2+}$ is required for

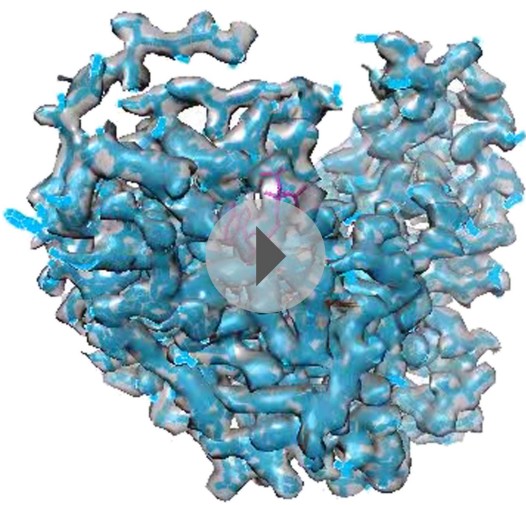

**Video 6.** GTase domain of t-CPV contains two ligands. The density map and atomic model of GTase domain in t-CPV are in transparent gray and sky blue, respectively. The density map is contoured at 3.0σ above the means. The atomic models of GTP and ATP are in orange and magenta, respectively. Mg$^{2+}$ is in green.

GTase activity of viruses in *Reoviridae* (*Yamakawa et al., 1982*; *Le Blois et al., 1992*; *Martinez-Costas et al., 1995*; *Qiu and Luongo, 2003*; *Mohd Jaafar et al., 2005*). Therefore, only the open state (with the putative Mg$^{2+}$) is catalytically active. Remarkably, the gate opening is achieved through the displacement of α14 (i.e., the gate helix), one of the three helices comprising the active site of the putative viral ATPase (*Figures 4A–C, 6B*). However, even though the gate helix α14 controls the open and closed state, it is not part of the GTase active site. In fact, the active sites of the putative ATPase and GTase do not share any amino acids or secondary elements (*Figure 1E*). Therefore, the putative ATPase regulates GTase activity allosterically.

The bound GTP molecules in the GTase open and closed states exhibit differences in their conformations and interactions with the GTase active site (*Figures 5A, 6*). The triphosphate moiety forms only one hydrogen bond (between the β phosphate and Tyr59) in the closed conformation (*Figure 5A*) but forms three or four more hydrogen bonds in the open conformation including the two formed by the α phosphate with His212 and Arg255 (*Figure 6A,C,D*). The more extensive hydrogen bonds observed in the open conformation is consistent with our assignment of it as the active state.

Catalysis of guanylyl transfer occurs in two steps: reaction with GTP to form a covalent enzyme–GMP intermediate (enzyme guanylylation) and transfer of GMP onto the 5′-diphosphorylated acceptor. Previous loss-of-function mutagenesis study of lysine residues in mammalian reovirus suggested that Lys190 of GTase domain is responsible for guanylylation of GTases (*Luongo et al., 2000*). Lys190 is located in a 28-aa segment (residues 168–195) that connects two structurally conserved β strands. Surprisingly, the conserved β strands (β3 and β4) in CPV GTPase is connected by a segment of only 13 aa (a loop from residues 166–178), which contains no lysine (*Figure 1D,E*). The connecting segments do not have sequence or structural similarities. Within the vicinity of the bound GTP, the only lysine residue in CPV GTase domain is Lys234, which maps to a non-conserved residue (Ser259) in mammalian reovirus GTase domain. Moreover, during GTase transition from its closed to open state, the α phosphorus of the GTP moves towards a histidine-rich segment and away from Lys234 (*Figures 6, 7* and *Figure 6—figure supplement 1*). Therefore, our structures indicate that Lys234 cannot directly participate in guanylylation of GTase, a conclusion that is contrary to a previous suggestion based on a likely incorrect placement of a GMP molecule in the active site in a poorer resolution map (*Yang et al., 2012*).

By contrast, the histidine-rich segment contains two histidines (His208 and His217) that are either hydrogen bonded to or in proximity with the α- or β-phosphate of the bound GTP in the open state of CPV GTase (*Figure 6A,C,D*) and conserved in both the orthoreovirus (*Reinisch et al., 2000*) and aquareovirus (*Zhang et al., 2010b*) (*Figure 6—figure supplement 2*). Located on the same side of the leaving group (i.e., β, γ-diphosphate), His208 is hydrogen bonded to the β-phosphate of the GTP in SGA-CPV and SG-CPV. Therefore, His208 is not the active site residue; rather, it stabilizes the charge built up on the β-phosphate in the transition state during the catalysis process. Instead, His217 is likely the active site residue. His217 and the leaving group are on opposite sides of the α phosphorus, a geometry suitable for in-line nucleophilic attack of the α phosphorus by His217 (*Figures 6, 7* and *Figure 6—figure supplement 2B*). Indeed, these two conserved histidine residues in orthoreovirus are required for the GTase activities (*Qiu and Luongo, 2003*). Furthermore, unlike KxDG GTases that have maximum activity at high pH, GTases of viruses in the *Reoviridae* family have maximum activity at pH about or lower than the pKa value (∼6.0) of histidine (*Qiu and Luongo, 2003*). Because we observed only the pre guanylylation state of GTase in all three CPV structures (t-CPV,

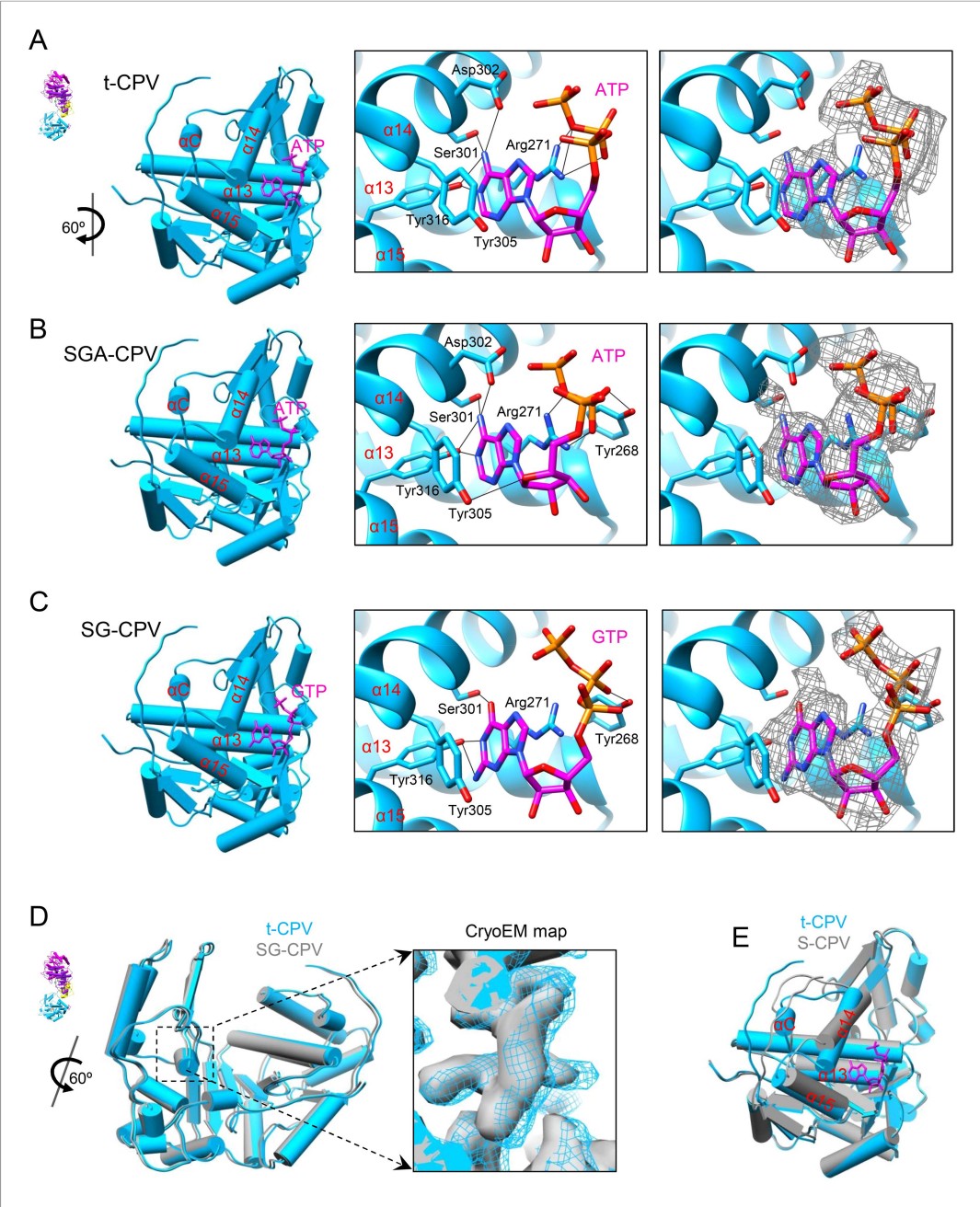

**Figure 4**. Discovery of the viral ATP-binding site. (**A**) Structure of GTase domain and ATP in t-CPV. Left, view rotated from the guide map (inset) as indicated. GTase domain is in sky blue. ATP is in magenta. Middle, zoom-in view of the putative ATP-binding site. ATP is colored by element: carbon atoms are magenta, nitrogen atoms are blue, and oxygen atoms are red. The hydrogen bonds are indicated by black lines. Side chains of Tyr305 and Arg271 form pi–pi and cation–pi interactions with the adenine ring of ATP, respectively. Right, same view as the middle. The density map of bound ATP (gray mesh) is contoured at 3σ above the means. (**B**) Structure of GTase domain and ATP in SGA-CPV. Molecules are viewed and colored as in **A**. (**C**) Structure of GTase domain and GTP in SG-CPV. Molecules are viewed and colored as in **A**. GTP is colored analogously. The density map of bound GTP (gray mesh) is contoured at 1.4σ above the means. (**D**) Superimposition of GTase domain between SG-CPV (gray) and t-CPV (sky blue). Inset: zoom-in view of the boxed region. Density maps from t-CPV (sky blue) and SG-CPV (gray) are contoured at 3.0σ above the means. (**E**) Superimposition of the large sub-domain of GTase domain between S-CPV (gray) and t-CPV (sky blue). Molecules were aligned using Cα positions for residues in small sub-domain. The bound ATP of t-CPV is in magenta.

*Figure 4. continued on next page*

*Figure 4. Continued*

The following figure supplements are available for figure 4:

**Figure supplement 1**. The pentameric turret complex of t-CPV.

**Figure supplement 2**. CryoEM of SGA-CPV.

**Figure supplement 3**. CryoEM of SG-CPV.

SGA-CPV, and SG-CPV), we reason that enzyme guanylylation mediated by His217 is likely the rate-limiting step in the process of guanylyl transfer.

Our structures also indicate that the putative viral ATPase regulates the methyl transfer activity of MT-1 through a long-range allosteric effect (*Figures 1D, 8* and *Figure 8—figure supplement 1*). First of all, active sites of MT-1 and the putative ATPase are spatially separated from each other (*Figure 1D* and *Figure 8—figure supplement 1*) as the distance from the putative ATPase site to the MT-1 in the same molecule is ∼80 Å, while that to a neighboring MT-1 is ∼40 Å (*Figure 8—figure supplement 1A*). Second, even if the putative ATPase is activated by SAM but lacks ATP for binding/hydrolysis, MT-1 remains incapable of SAM binding as was observed in S-CPV. In t-CPV where ATP is available, MT-1 becomes SAM bound (*Figure 8* and *Figure 8—figure supplement 2A*). The structures of MT-2 in S-CPV and t-CPV are essentially identical (*Figure 8A,B*), but their MT-1 structures differ. In particular, two loops lining one side of the un-occupied MT-1 active site in S-CPV shifted up to 4 Å in t-CPV, resulting in an enlarged active site to accommodate the SAM molecule required for methyl transfer (*Figure 8C,D*). In SGA-CPV and SG-CPV, the structures of MT-1 domains with bound SAM are essentially identical to those in t-CPV (*Figure 8—figure supplement 2*), though the outwards movement of MT-1 in SG-CPV is ∼1.1 Å less (*Figure 8E*), likely due to the lower rate of GTP hydrolysis by the ATPase (*Figure 5B*).

## Discussion

In this study, we discovered that the large sub-domain of CPV GTase domain has an ATP-binding site and is likely an ATPase. This putative viral ATPase has the conserved structural motif for recognition of the adenine base of ATP: hydrogen bonds between the side chains of Ser301, Asp302 and Tyr316 and the adenine

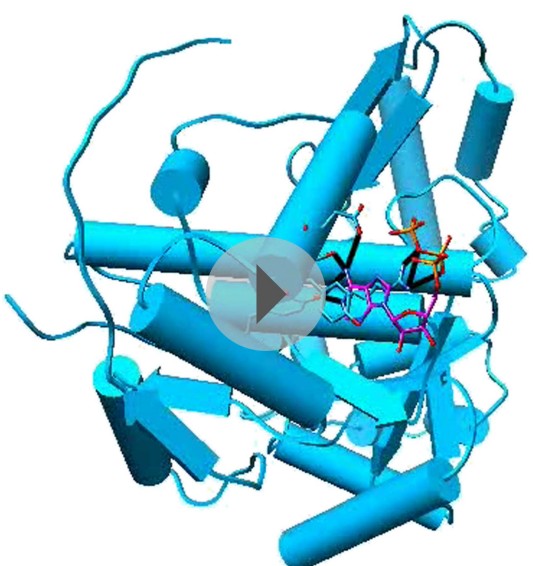

**Video 7.** Structure of ATP-binding site and the bound ATP in t-CPV. The atomic model of GTase domain is in sky blue. ATP is colored by element as in *Figure 4A*. Side chains of amino acid involved in hydrogen bonding (black lines) or stacking with ATP are shown. The density map of the bound ATP is contoured at 3.0σ above the means.

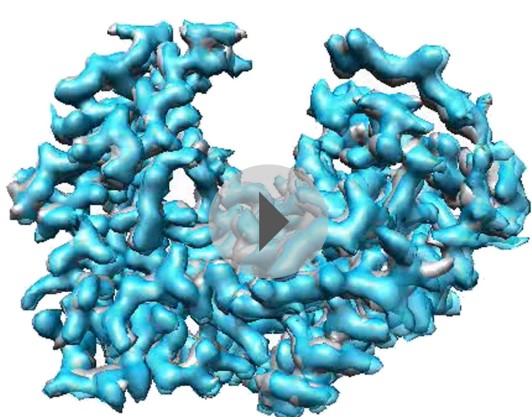

**Video 8.** Structure comparison of GTase domain between SG-CPV (gray) and t-CPV (sky blue).

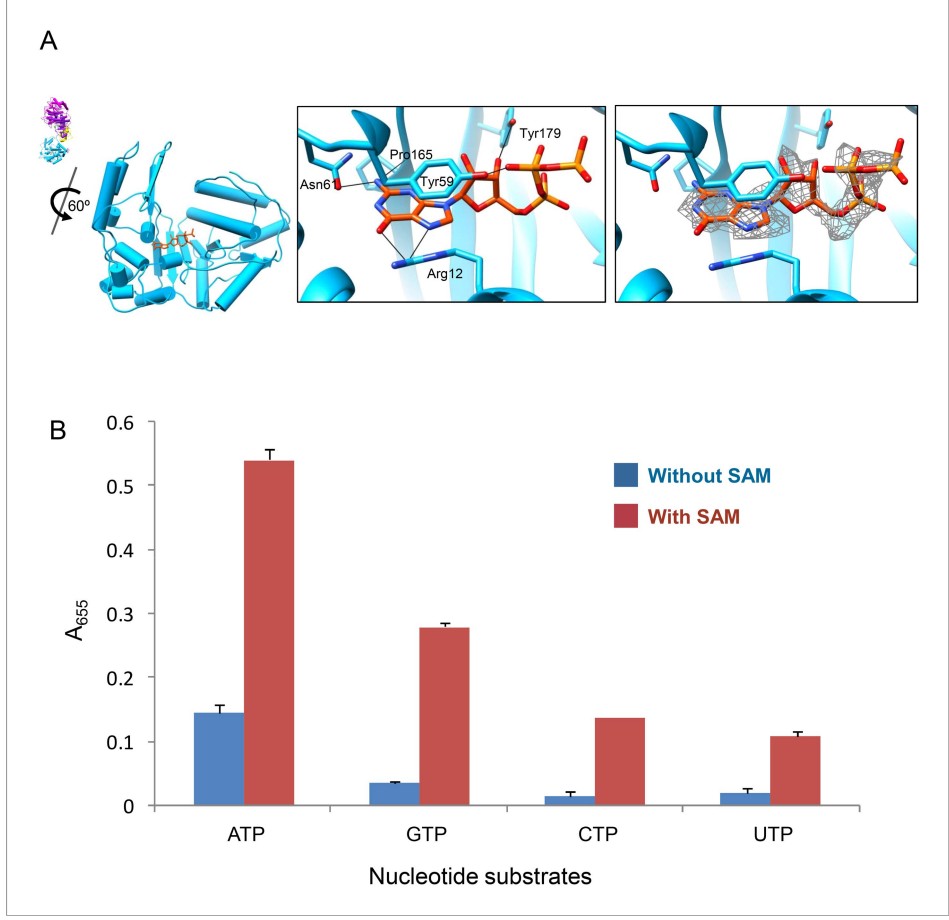

**Figure 5**. ATP binding and hydrolysis by the viral ATPase is SAM-dependent. (**A**) Structure of GTase domain and GTP in G-CPV. Left, view rotated from the guide map (inset) as indicated. GTase domain is in sky blue. GTP is in orange red. Middle, active site of GTase. GTP is colored by element: carbon atoms are orange red, nitrogen atoms are blue, and oxygen atoms are red. The hydrogen bonds are indicated by black lines. Side chain of Tyr59 also forms pi–pi stacking interaction with the guanylyl ring of GTP. Right, same view as the middle. The density map of bound GTP (gray mesh) is contoured at 3σ above the means. (**B**) Nucleotide substrates specificity by CPV nucleoside triphosphatase. Values are means derived from duplicate experiments. Standard deviations are indicated by error bar.

The following figure supplement is available for figure 5:

**Figure supplement 1**. Stereo view of GTPase site and GTP in G-CPV.

ring, pi–pi stacking between the side chain of Tyr305 with the adenine base, and the cation–pi stacking between the side chain of Arg271 and the adenine base (*Figure 4A,B*). However, this putative viral ATPase lacks the structural motifs of canonical nucleoside triphosphatases (NTPases) (including cellular kinases), most notably the P loop, for binding the phosphoryl moiety of NTP (*Saraste et al., 1990*; *Smith and Rayment, 1996*; *Snider and Houry, 2008*). Instead, the phosphoryl group of the bound ATP is stabilized through hydrogen bonds with the side chain of Arg271 or Tyr268 from α13 helix (*Figure 4A,B*). Although the large sub-domain has an α helices/β sheet fold, the putative viral ATPase active site is composed of three consecutive helices of α13, α14, and α15 (*Figure 4A,B*). Structurally, the putative ATPase is different from all ATPase known to date, and it may thus represent a new type of ATPase.

Although both GTP and ATP can bind at the putative viral ATPase site, their interactions with the protein have some differences (*Figure 4A–C*). The base and ribose of the bound GTP are less hydrogen bounded to the active site than those of the bound ATP. More importantly, while the

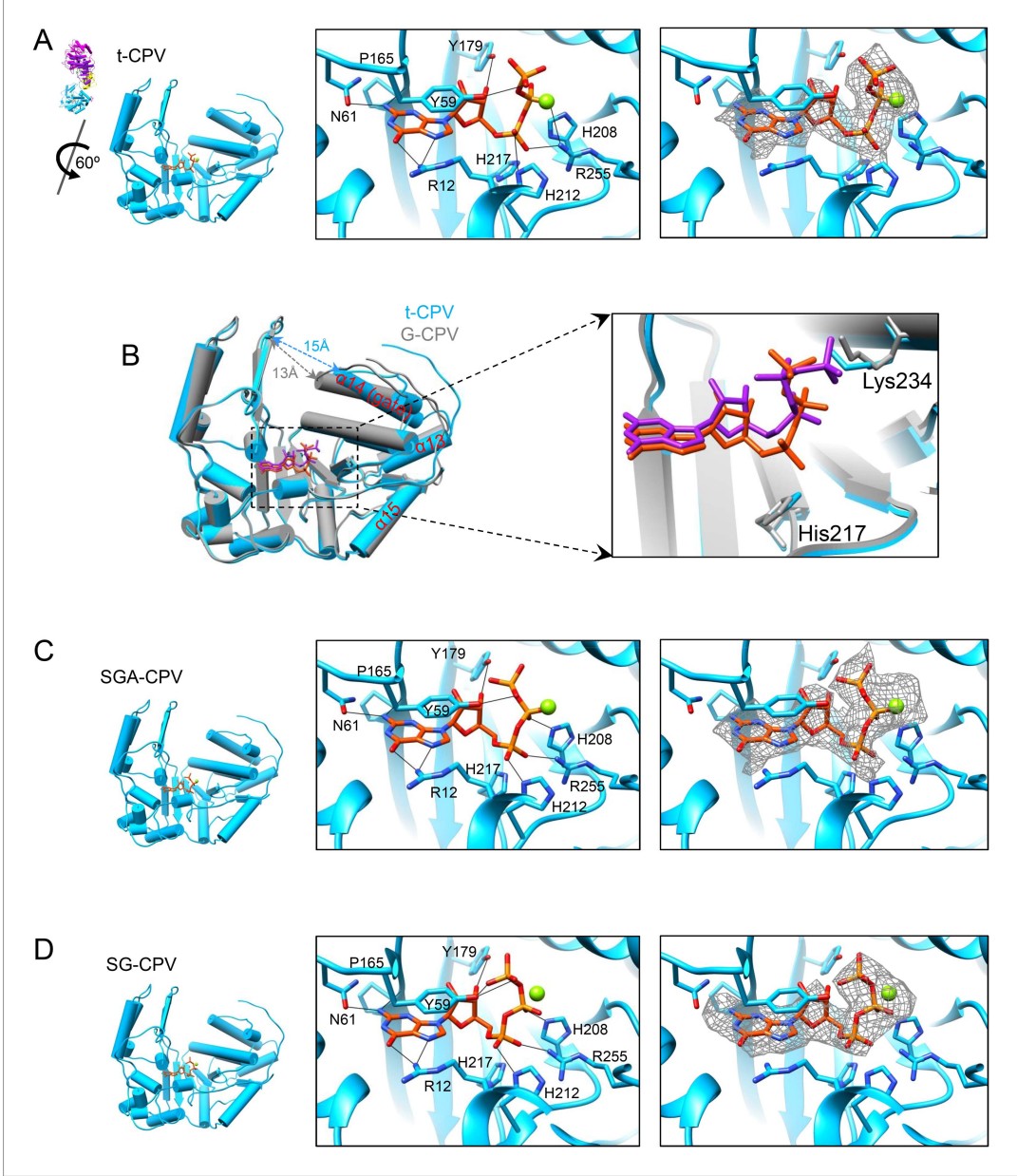

**Figure 6.** The catalytic activity of viral GTase is regulated by the viral ATPase through allosteric effect. (**A**) Structure of GTase domain and the bound $Mg^{2+}$-GTP in t-CPV. Left, view rotated from the guide map (inset) as indicated. GTase domain is in sky blue. GTP is in orange red. $Mg^{2+}$ is in green. Middle, active site of GTase with bound $Mg^{2+}$-GTP. GTP is colored by element as in **Figure 5A**. The hydrogen bonds are indicated by black lines. Side chains of the two conserved His208 and His217 are shown. Right, same view as the middle. The density map of bound GTP (gray mesh) is contoured at 3σ above the means. (**B**) Superimposition of GTase domain between G-CPV (gray) and t-CPV (sky blue). Molecules were aligned using Ca positions for residues in small sub-domain. GTPs bound to the GTase sites of G-CPV and t-CPV are in purple and orange red, respectively. Inset, zoom-in view of the boxed region. (**C**) Structure of GTase domain and the bound $Mg^{2+}$-GTP in SGA-CPV. Molecules and $Mg^{2+}$ are viewed and colored as in **A**. Side chain of the conserved His217 is shown. (**D**) Structure of GTase domain and the bound $Mg^{2+}$-GTP in SG-CPV. Molecules and $Mg^{2+}$ are shown as in **A**. Side chain of the conserved His217 is shown.

The following figure supplements are available for figure 6:

**Figure supplement 1**. Structures of GTase sites and bound GTPs.

**Figure supplement 2**. The conserved His217 is the catalytic amino acid for guanylylation of GTase in CPV.

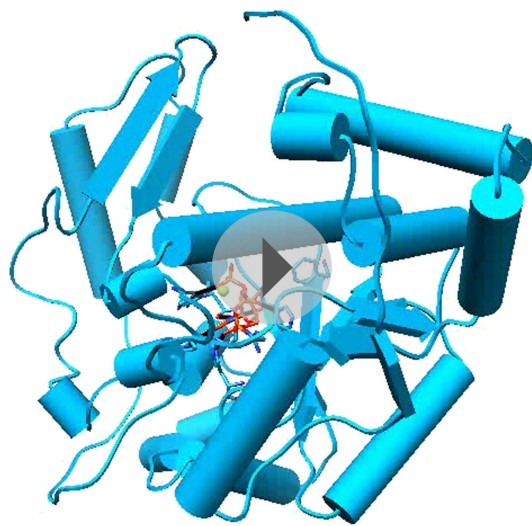

**Video 9.** Structure of GTase active site and the bound Mg-GTP in SGA-CPV. Color coding: sky blue—atomic model of GTase domain; green—Mg$^{2+}$; GTP—colored by element as in *Figure 6C*. Side chains of amino acids of the conserved His217, and those involved in hydrogen bonding or in stacking with GTP are shown. The density map of the bound GTP is shown as mesh at a contour level of 3.0σ above the means.

triphosphate group of the bound GTP forms only one hydrogen bond through its β phosphate with Tyr268 of the active site (*Figure 4C*), the triphosphate moiety of the bound ATP is better stabilized by forming two more hydrogen bonds with the protein (*Figure 4A,B*). Conceivably, the more hydrogen bonded ATP is a more efficient substrate of the putative viral ATPase for hydrolysis than GTP, consistent with the colorimetric assay of CPV NTPase activity (*Figure 5B*).

By integrating the atomic structures of six different CPV particles and correlation with NTPase assay results, we propose a viral ATPase-mediated activation of mRNA transcription and capping, as illustrated in *Figure 9*. As a virus must rely on host cell for replication, it is to the best interest of the virus to remain quiescent outside host cells (*Figure 9A*). CPV senses the entrance into host cytoplasm by detecting the presence of SAM. SAM, acting as a signal and binding to its receptor of MT-2, causes initial conformational change of the virus capsid, which activates the putative viral ATPase (*Figure 9B*). The activated viral ATPase then binds and hydrolyzes ATP to cause three major structural transformations, leading to mRNA transcription and capping (*Figure 9C*). First, as a result of the translocation of CSP, the viral capsid is enlarged, facilitating dsRNA template movement and enabling efficient mRNA synthesis (*Figure 9C*). Second, the GTase domain transforms from its closed to open state (*Figure 9C*). Although GTP binds to the GTase active site in both states of GTase domain, only in the open state can the GTase bind Mg$^{2+}$ and catalyze His217-mediated guanylyl transfer. Third, the MT-1 domain transforms from its closed to open conformation (*Figure 9C*). Only in its open conformation can MT-1 bind SAM. While the MT-1 and GTase domains from the same molecule are separated by the bridge domain and have no direct contact with each other, the MT-1 from a neighboring TP sits atop the putative ATPase sub-domain of GTase domain (*Figure 8—figure supplement 1*). We, therefore propose that the putative ATPase regulates the activity of MT-1 in a neighboring TP, probably through the conformational changes of the putative ATPase sub-domain upon ATP binding/hydrolysis. Notably, from S-CPV to t-CPV, the C-terminal loop of GTase domain exhibits significant movement towards the MT-1 domain of its neighboring TP molecule, presumably contributing to open the active site of the MT-1 (*Figure 8—figure supplement 1*).

Several viruses within the *Reoviridae*, such as rotaviruses and blue-tongue viruses, cause wide spread diseases in human and live stocks. Some of these multi-shelled viruses in the *Reoviridae*, including the animal reovirus and blue-tongue virus, have been shown to have ATPase activity (*Noble and Nibert, 1997*; *Ramadevi and Roy, 1998*). The remarkable parallel between the ATPase activity and the transcription activity indicated that they may have also employed the ATPase-mediated activation for mRNA transcription and capping identified here. For example, reovirus cores had high level of ATPase activity (*Noble and Nibert, 1997*) and could synthesize mRNA (*Shatkin and Sipe, 1968*; *Banerjee and Shatkin, 1970*; *Drayna and Fields, 1982*; *Farsetta et al., 2000*), but virions had little ATPase activity (*Noble and Nibert, 1997*) and could not synthesize mRNA (*Shatkin and Sipe, 1968*; *Farsetta et al., 2000*). Thus, it is the removal of outer shell other than SAM that triggers the activation process in multi-shelled reoviruses.

## Materials and methods

### Sample preparation
CPV virions were purified as described (*Yu et al., 2008*). Briefly, purified polyhedra were treated with an alkaline solution of 0.2 M Na$_2$CO$_3$-NaHCO$_3$ (pH 10.8) for 1 hr. The suspension was centrifuged at

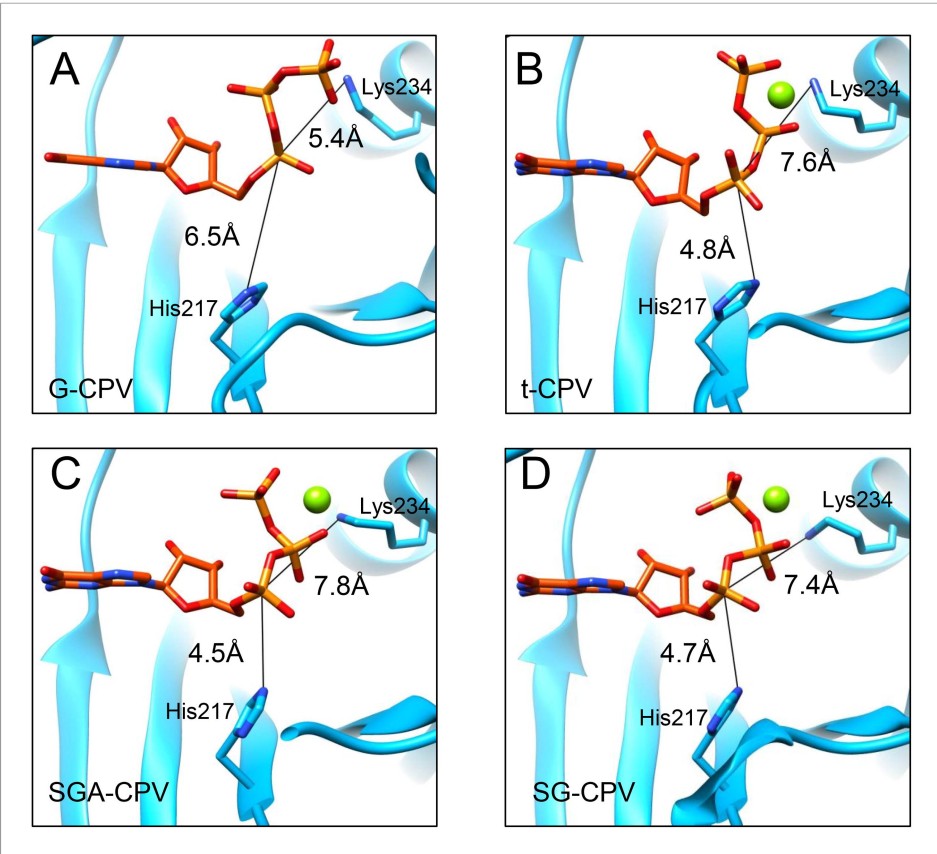

**Figure 7**. The α-phosphorus of GTP bound to the GTase site moves towards His217 and away from Lys234 accompanying the activation of GTase. (**A**) The distance between the Nε2 of His217 and the α-phosphorus of GTP in G-CPV is ~6.5 Å. The distance between the Nε of Lys234 and the α-phosphorus is ~5.4 Å. Molecules and $Mg^{2+}$ are colored as in *Figure 5B*. (**B**) The distance between Nε2 of His217 and the α-phosphorus of GTP in t-CPV is ~4.8 Å. The distance between the Nε of Lys234 and the α-phosphorus is ~7.6 Å. (**C**) The distance between Nε2 of His217 and the α-phosphorus of GTP in SGA-CPV is ~4.5 Å. The distance between the Nε of Lys234 and the α-phosphorus is ~7.8 Å. (**D**) The distance between Nε2 of His217 and the α-phosphorus of GTP in SG-CPV is ~4.7 Å. The distance between the Nε of Lys234 and the α-phosphorus is ~7.4 Å.

10,000×*g* for 40 min. The resulting supernatant was collected and then centrifuged again at 80,000×*g* for 60 min at 4°C to pellet the CPV virions. The final pellet was re-suspended in a reaction buffer (70 mM pH 8.0 Tris-Cl, 10 mM $MgCl_2$, and 100 mM NaCl) and used for the following experiments.

We prepared six different CPV samples using a protocol modified from a previously described CPV transcription essay (*Smith and Furuichi, 1980*). Reaction mixtures (30 µl) contained purified CPV, 70 mM Tris-Cl (pH 8.0), 10 mM $MgCl_2$, 100 mM NaCl, and 1 mM SAM (S-CPV), or 1 mM SAM+2 mM GTP+2 mM UTP+2 mM CTP+4 mM ATP (t-CPV), or 1 mM SAM+2 mM GTP+ 2 mM ATP (SGA-CPV), or 1 mM SAM+2 mM GTP (SG-CPV), or 2 mM GTP (G-CPV), or 2 mM ATP (A-CPV). All reactions were incubated at 31°C for 15 min and stopped by quenching the reaction tubes on ice.

## CryoEM imaging and 3D reconstruction

Each of the six different CPV particles mentioned above was embedded in a thin layer of vitreous ice suspended across the holes of holey carbon films by plunge-freezing into liquid ethane. Before data collection, beam tilt was carefully minimized by coma-free alignment. Viral particle samples were kept at liquid-nitrogen temperature. CryoEM images were recorded on Kodak SO163 films at a dosage of ~25 electrons/Å2 on an FEI Titan Krios cryo-electron microscope operated at 300 kV and 59,000× nominal magnification with parallel beam illumination. The films were digitized with a Nikon scanner at

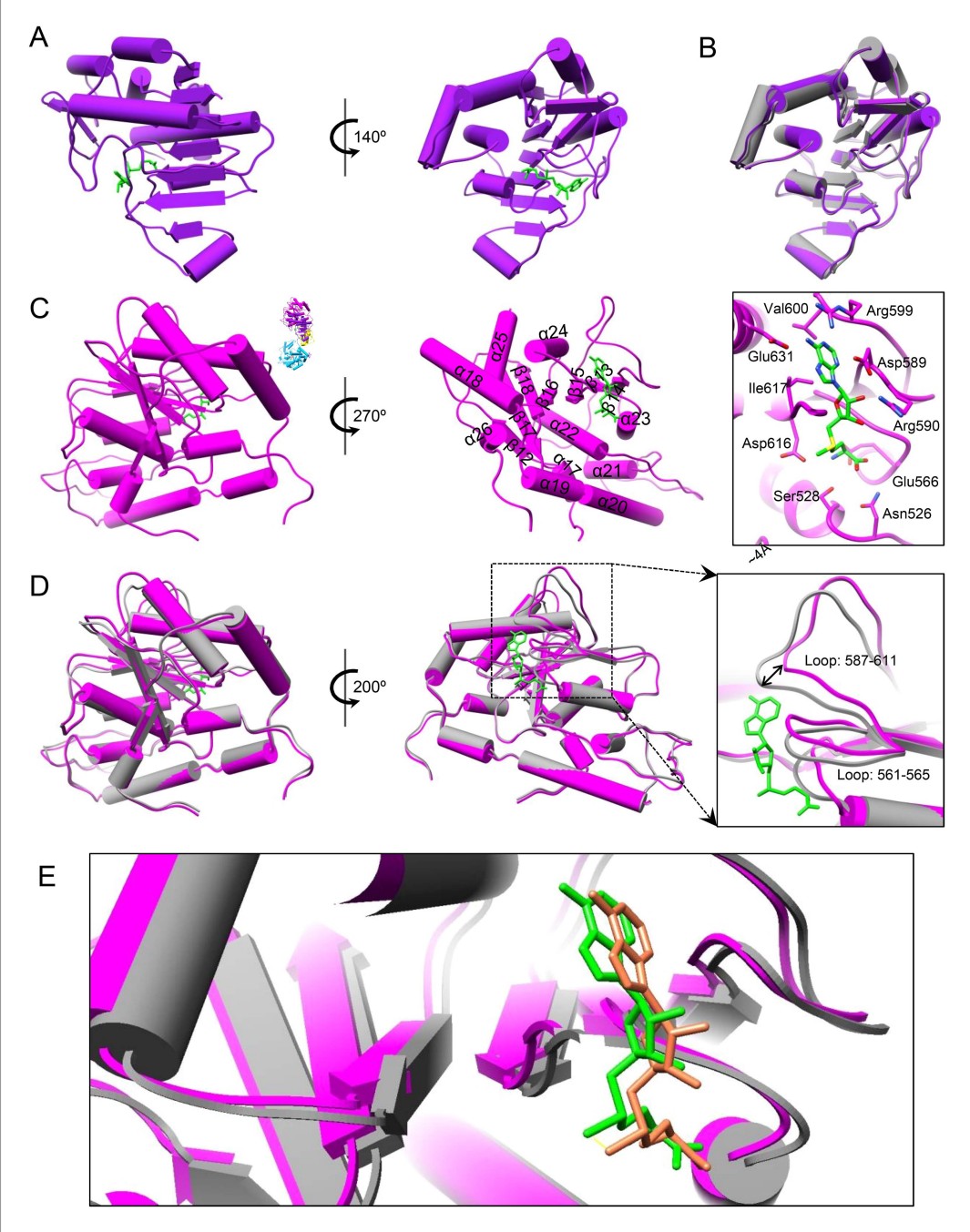

**Figure 8**. The catalytic activity of MT-1 is also regulated by the viral ATPase through allosteric effect. (**A**) Structure of MT-2 domain and the bound SAM in t-CPV. MT-2 domain is in purple. SAM is in green. Left, viewed as in *Figure 2C*. Right, view rotated as indicated. (**B**) Superimposition of MT-2 between S-CPV (gray) and t-CPV (purple). Molecules were aligned using Ca positions for residues in domain. (**C**) Structure of MT-1 domain and the bound SAM in t-CPV. MT-1 domain is in magenta. SAM is in green. Left, view as the guide map (inset). Middle, view rotated as indicated. Right, active site of MT-1 domain. SAM is colored as in *Figure 2D*. Side chains of amino acids involved in interactions with SAM are shown. (**D**) Superimposition of MT-1 between S-CPV (gray) and t-CPV (magenta). Molecules were aligned using Ca positions for residues in MT-1 domain. The bound SAM of t-CPV is in green. Left, viewed as the guide map in **C**. Right, view rotated as indicated. Inset: zoom-in view of MT-1 active site. (**E**) Superimposition of MT-1 active site between SG-CPV (gray) and t-CPV (magenta). The SAM molecules bound to the active sites of SG-CPV and t-CPV are colored in coral and green, respectively.

*Figure 8. continued on next page*

*Figure 8. Continued*

The following figure supplements are available for figure 8:

**Figure supplement 1**. The putative viral ATPase regulates the methyl transfer activity of MT-1.

**Figure supplement 2**. Structures of MT-1 active sites and SAMs.

a step size of 6.35 µm/pixel, corresponding to 1.076 Å/pixel at the sample level. Individual particle images (960 × 960 pixels) were first boxed out automatically by the *autoBox* program in the IMIRS package (*Liang et al., 2002*) and then followed by manual screening using the EMAN *boxer* program (*Ludtke et al., 1999*) to keep only the well-separated, contamination-free, intact RNA-containing particles.

The program *CTFFIND* (*Mindell and Grigorieff, 2003*) was used to determine the defocus value and astigmatism parameters for each micrograph. We determined particle orientation, center parameters with the IMIRS package running in MPI-enabled Windows workstations (*Liang et al., 2002*). 3D reconstruction was performed by *eLite3D* using graphical processing units (*Zhang et al., 2010a*). We considered astigmatism during CTF correction in the orientation/center refinement and 3D reconstruction steps.

Effective resolutions of the final reconstructions were estimated to be 2.9–3.1 Å (*Figure 1A–C*, *Table 1* and *Figure 1—figure supplement 1*), based on the structural features revealed in the cryoEM density maps, R-factors (*Wolf et al., 2010*), and Fourier shell correlation coefficient (FSC) criterion as defined by Rosenthal and Henderson (*Rosenthal and Henderson, 2003*). We have previously shown that our common-lines-based programs do not suffer from the problems of over-fitting or model bias (*Zhou et al., 2014*). To provide further validation, we took advantage of the existence of identical structures of t-CPV and SGA-CPV (both independently determined) and calculated the FSC curves between the SGA-CPV map and SGA-CPV model and that between the SGA-CPV map and t-CPV model. Because these structures were independently determined, they are essentially the same as 'gold-standard' FSC (*Scheres and Chen, 2012*). These analyses further support our conclusion that our reconstructions do not have over-fitting (*Figure 1—figure supplement 1B*).

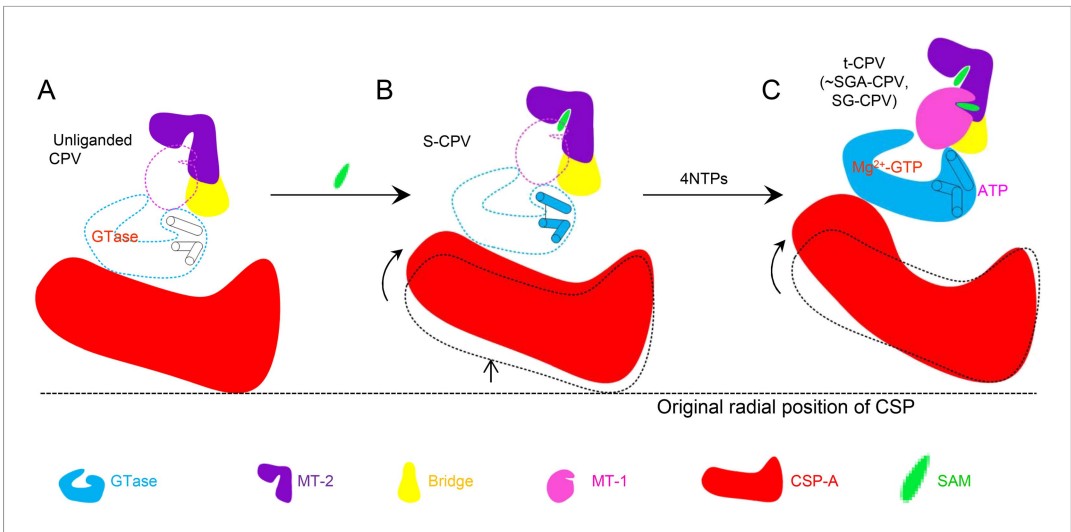

**Figure 9**. Schematic illustration of the putative viral ATPase-mediated activation of mRNA transcription and capping. In this illustration, the active open states of enzymes are shown in filled colors and the inactive closed states of enzymes are shown in dotted color lines. (**A**) CSP-A (red) and GTase, bridge and MT-2 domains from the same TP molecule, and a neighboring MT-1 (colored as in *Figure 1D*) of unliganded CPV. The inactive ATPase site is indicated by three empty cylinders. (**B**) CSP-A and GTase, bridge and MT-2 domains from the same TP molecule, and a neighboring MT-1of S-CPV. SAM alone can only bind to MT-2 domain (purple) to cause conformational change and activate the putative viral ATPase. The activated ATPase site is indicated by three colored cylinders. (**C**) CSP-A and GTase, bridge and MT-2 domains from the same molecule, and a neighboring MT-1 of t-CPV.

## Atomic model building, model refinement, and 3D visualization

Rebuilding the model to fit the EM map was done manually with *COOT* (*Emsley and Cowtan, 2004*) with the help of *REMO* (*Li and Zhang, 2009*). The 'regularize zone' utility of *COOT* was used to improve model stereochemistry.

These coarse full-atom models were then refined in a pseudocrystallographic manner using Phenix (*Adams et al., 2010*). This procedure only improves atomic models and does not modify the cryoEM density map. Densities for individual proteins were segmented, put in artificial crystal lattices, and then used to calculate their structural factors. The amplitudes and phases of these structural factors were used as pseudo-experimental diffraction data for model refinement in Phenix. To improve the areas of interaction between different protein subunits, we put the refined structures of all five subunits from an asymmetric unit into a single coordinate file and pseudo-crystallographically refined them simultaneously with their non-crystallographic symmetry. This refinement process uses pseudo-experimental diffraction data generated from the cryoEM map of an asymmetric unit.

CryoEM reconstruction was visualized and segmented using *Chimera* (*Pettersen et al., 2004*). All figures were prepared with *Chimera* and *COOT*.

## Determination of CPV NTPase activity by colorimetric assay

The NTPase reactions and colorimetric assay were performed as described by *Noble and Nibert (1997)* in 1.5-ml eppendorf tubes. NTPase reaction mixtures contained 100 mM Tris-Cl (pH 8.0), 100 mM NaCl, 10 mM $MgCl_2$, without or with 1 mM SAM, $6 \times 10^{11}$ CPV particles per ml, and 1 mM of one of the 4 NTPs in a total volume of 60 µl. Reaction components were mixed on ice, incubated at 31°C for 30 min and then returned to ice. Termination of each reaction was ensured by the addition of an equal volume of 10% trichloroacetic acid. To measure the amount of phosphate ion in each sample, the stopped reaction mixture was mixed with an equal volume of colorimetric reagent (3 vol of 0.8% ammonium molybdate, 1 vol of 6 N sulfuric acid, 1 vol of 10% [wt/vol] ascorbic acid). After all samples in the experiment were added, the eppendorf tubes were incubated in a water bath at 31°C for 30 min. During development, a reduced phosphomolybdate complex was formed, which was blue in color and quantifiable by $A_{655}$ nm. In each experiment, samples containing NTP but no CPV were included to permit correction for background.

## Accession numbers

The cryoEM density maps and atomic coordinates reported here are deposited in the EM Data Bank and the Protein Data Bank with accession codes EMD-6371 (A-CPV), EMD-6374 (G-CPV), EMD-6375 (S-CPV), EMD-6376 (SG-CPV), EMD-6377 (SGA-CPV), EMD-6378 (t-CPV) (*Yu et al., 2015a*; *2015b*; *2015c*; *2015d*; *2015e*; *2015f*) and 3JAZ (A-CPV), 3JB0 (G-CPV), 3JB1 (S-CPV), 3JB2 (SG-CPV), 3JB3 (SGA-CPV), 3JAY (t-CPV) (*Yu et al., 2015g*; *2015h*; *2015i*; *2015j*; *2015k*; *2015l*), respectively.

## Acknowledgements

We thank Xing Zhang for help in model refinement, Jonathan Jih for assistance in model building, and Ke Ding for making *Figure 9*. This research is supported in part by the National Institute of Health (GM071940 and AI094386 to ZHZ), the National Natural Science Foundation of China (31172263 to JS), the Natural Science Foundation of Guangdong Province, China (S2013010016750 to JS), and Specialized Research Fund for the Doctoral Program of Higher Education, China (20134404110023 to JS). We acknowledge the use of instruments at the Electron Imaging Center for Nanomachines supported by UCLA and by instrumentation grants from NIH (1S10RR23057) and NSF (DBI-1338135).

## Additional information

### Funding

| Funder | Grant reference | Author |
| --- | --- | --- |
| National Institutes of Health (NIH) | GM071940 | Z Hong Zhou |
| National Natural Science Foundation of China (NSFC) | 31172263 | Jingchen Sun |
| National Institutes of Health (NIH) | AI094386 | Z Hong Zhou |

| Funder | Grant reference | Author |
| --- | --- | --- |
| Natural Science Foundation of Guangdong Province | S2013010016750 | Jingchen Sun |
| Specialized Research Fund for the Doctoral Program of Higher Education of China (SRFDP) | 20134404110023 | Jingchen Sun |

The funders had no role in study design, data collection and interpretation, or the decision to submit the work for publication.

## Author contributions

XY, Conception and design, Acquisition of data and performed the NTPase assay, Analysis and interpretation of data, Drafting or revising the article; JJ, Recorded cryoEM images, Performed the NTPase assay and critically reading the final paper, Acquisition of data; JS, Provided polyhedra and performed the NTPase assay; ZHZ, Conception and design, Analysis and interpretation of data, Drafting or revising the article

# Additional files

## Major datasets

The following datasets were generated:

| Author(s) | Year | Dataset title | Dataset ID and/or URL | Database, license, and accessibility information |
| --- | --- | --- | --- | --- |
| Yu X, Jiang J, Sun J, Zhou ZH | 2015 | A putative ATPase mediates RNA transcription and capping in a dsRNA virus | http://www.rcsb.org/pdb/search/structidSearch.do?structureId=3JAZ | Publicly available at the RCSB Protein Data Bank (accession number 3JAZ). |
| Yu X, Jiang J, Sun J, Zhou ZH | 2015 | A putative ATPase mediates RNA transcription and capping in a dsRNA virus | http://www.rcsb.org/pdb/search/structidSearch.do?structureId=3JB0 | Publicly available at the RCSB Protein Data Bank (accession number 3JB0). |
| Yu X, Jiang J, Sun J, Zhou ZH | 2015 | A putative ATPase mediates RNA transcription and capping in a dsRNA virus | http://www.rcsb.org/pdb/search/structidSearch.do?structureId=3JB1 | Publicly available at the RCSB Protein Data Bank (accession number 3JB1). |
| Yu X, Jiang J, Sun J, Zhou ZH | 2015 | A putative ATPase mediates RNA transcription and capping in a dsRNA virus | http://www.rcsb.org/pdb/search/structidSearch.do?structureId=3JB2 | Publicly available at the RCSB Protein Data Bank (accession number 3JB2). |
| Yu X, Jiang J, Sun J, Zhou ZH | 2015 | A putative ATPase mediates RNA transcription and capping in a dsRNA virus | http://www.rcsb.org/pdb/search/structidSearch.do?structureId=3JB3 | Publicly available at the RCSB Protein Data Bank (accession number 3JB3). |
| Yu X, Jiang J, Sun J, Zhou ZH | 2015 | A putative ATPase mediates RNA transcription and capping in a dsRNA virus | http://www.rcsb.org/pdb/search/structidSearch.do?structureId=3JAY | Publicly available at the RCSB Protein Data Bank (accession number 3JAY). |
| Yu X, Jiang J, Sun J, Zhou ZH | 2015 | A putative ATPase mediates RNA transcription and capping in a dsRNA virus | http://www.ebi.ac.uk/pdbe/entry/emdb/EMD-6371 | Publicly available at the Electron Microscopy Data Bank (accession number EMD-6371). |
| Yu X, Jiang J, Sun J, Zhou ZH | 2015 | A putative ATPase mediates RNA transcription and capping in a dsRNA virus | http://www.ebi.ac.uk/pdbe/entry/emdb/EMD-6374 | Publicly available at the Electron Microscopy Data Bank (accession number EMD-6374). |
| Yu X, Jiang J, Sun J, Zhou ZH | 2015 | A putative ATPase mediates RNA transcription and capping in a dsRNA virus | http://www.ebi.ac.uk/pdbe/entry/emdb/EMD-6375 | Publicly available at the Electron Microscopy Data Bank (accession number EMD-6375). |

| Author(s) | Year | Dataset title | Dataset ID and/or URL | Database, license, and accessibility information |
|---|---|---|---|---|
| Yu X, Jiang J, Sun J, Zhou ZH | 2015 | A putative ATPase mediates RNA transcription and capping in a dsRNA virus | http://www.ebi.ac.uk/pdbe/entry/emdb/EMD-6376 | Publicly available at the Electron Microscopy Data Bank (accession number EMD-6376). |
| Yu X, Jiang J, Sun J, Zhou ZH | 2015 | A putative ATPase mediates RNA transcription and capping in a dsRNA virus | http://www.ebi.ac.uk/pdbe/entry/emdb/EMD-6377 | Publicly available at the Electron Microscopy Data Bank (accession number EMD-6377). |
| Yu X, Jiang J, Sun J, Zhou ZH | 2015 | A putative ATPase mediates RNA transcription and capping in a dsRNA virus | http://www.ebi.ac.uk/pdbe/entry/emdb/EMD-6378 | Publicly available at the Electron Microscopy Data Bank (accession number EMD-6378). |

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
