## [Decision Letter]

Thank you for sending your work entitled “ATP-binding mediates RNA transcription and capping in a dsRNA virus” for consideration at *eLife*. Your article has been favorably evaluated by Michael Marletta (Senior editor) and three reviewers, one of whom is a member of our Board of Reviewing Editors. The following individuals responsible for the peer review of your submission have agreed to reveal their identity: Sjors Scheres (Reviewing editor) and Venkatar Prasad (peer reviewer). A further reviewer remains anonymous.

All three reviewers thought that the structures presented were of excellent quality and represented a tour-de-force use of near atomic resolution cryo-EM on fairly complicated viral system to derive the mechanistic basis of an intrinsic feature that is common to turreted reoviruses. In particular, the mechanism of SAM activation of the transcriptase was considered to be a nice contribution. Still, a few concerns were raised that would require your careful attention in a revised manuscript. These are (in order of importance):

1) The suggested ATPase activity at the observed site and its postulated role during transcription are not convincing. The presence of ATP bound there where they see it does not mean that this is the ATPase active site. Normally, to catch ATP bound at the active site of an ATPase, it is necessary to use a non-hydrolysable analog. Your manuscript shows also that the particle hydrolyses ATP in the presence of SAM, so why would this ATP molecule be intact in the ATPase active site? In addition, ATP is hydrolyzed usually to recover the energy for some other function, like in motor proteins. It is not obvious here how is the energy of ATP hydrolysis used during transcription. Would its hydrolysis lead to a further change of conformation of the TP? Or how do you envision the use of the ATPase to help in transcription? Currently, the data do not justify the title “ATP-binding mediates RNA transcription and capping in a dsRNA virus”.

2) The description of the actual conformational changes is difficult to follow, and at points appears contradictory. The figures are also not very helpful in showing the conformational change. Morphings of the particle from one state to the others, provided as supplementary animations, would help a lot.

3) Also, the observed conformational differences among some of these maps (e.g. S-CPV and CPV) are very small to support the hypothesis that 'the enlarged capsid may facilitate dsRNA template movement'. Could small differences in magnification have occurred among the different data sets and provide an alternative explanation for the enlarged capsid size? Could you also provide an estimate of the internal cavity in cubic Angstroms for each of the relevant reconstructions?

4) It is not clear how the location of the magnesium location is identified. Whether the typical magnesium coordination geometry is satisfied, and how tentative is the assignment needs to be addressed.

5) In the t-CPV structure: it looks like the transcripts are getting extruded, but it appears there is no density that is observed in the turrets that can be assigned to exiting RNA molecules. Considering that endogenous transcription is a dynamic process, one would expect some of the structural features in the exit pathway should be smeared; however, there is no discussion of this in the manuscript.

The density in the presented figures and movie looks quite good, so the following comments may not be that relevant to this particular paper. However, it is good practice to show clarity in all manuscripts about how potential overfitting was assessed in both cryo-EM map refinement and the atomic model refinement inside that map.

6) You state that 'We have previously shown that our common-lines based programs do not suffer from the problems of over-fitting or model bias (47)'. However, the [47] paper doesn't explicitly mention overfitting, although it does show a gold-standard FSC curve in the supplementary information. Does that mean that you used gold-standard refinement procedures in the calculation of these maps? If so, you should mention this explicitly instead of citing the [47] paper. If you didn't, then refinements should be redone in the gold-standard manner (or by limiting the resolution used in refinement). Also, it would be good to show (gold-standard) FSC curves for each reconstruction in Figure 1—figure supplement 1.

7) R-factors are reported for the refined atomic models. Also in this refinement overfitting can be an issue. Without proper cross-validation, the R-factor curves in Figure 1—figure supplement 1 cannot be trusted. You could either choose to limit the resolution used in the model refinement (and then assess the falloff of the R-factor or the FSC between the model and the map beyond that resolution, i.e. the predictive power of the model), or you could refine a scrambled version of the model in one of the gold-standard half-maps and then calculate FSC/R-factors between that model and that same half-map as well as FSC/R-factors between that model and the other half-map. The difference between these 2 curves is an indication of the amount of overfitting in the model. In the absence of overfitting, the 2 curves should overlap.

---

## [Author Response]

*1) The suggested ATPase activity at the observed site and its postulated role during transcription are not convincing. The presence of ATP bound there where they see it does not mean that this is the ATPase active site. Normally, to catch ATP bound at the active site of an ATPase, it is necessary to use a non-hydrolysable analog. Your manuscript shows also that the particle hydrolyses ATP in the presence of SAM, so why would this ATP molecule be intact in the ATPase active site? In addition, ATP is hydrolyzed usually to recover the energy for some other function, like in motor proteins*.

An intact ATP can be observed at the active site when ATP hydrolysis at that site is a slow, rate-limiting step. Once ATP is hydrolyzed to ADP, it would be rapidly replaced by a new ATP. For the same reason, we only observed GTP—not the GMP product—at the active site of CPV GTase. We have performed preliminary reconstruction with an ATP analog (AMP-PNP) and indeed, the analog was observed at the expected site (see Figure 10).

Author response image 1.CryoEM of SGnA-CPV. GTase domain and bound AMP-PNP in SGnA-CPV (CPV+SAM+GTP+non-hydrosable AMP-PNP). ATP model is colored by element. The density map and atomic model of GTase are in gray and sky blue, respectively. Density map is contoured at 4σ above the means.**DOI:**
http://dx.doi.org/10.7554/eLife.07901.035

It is not obvious here how is the energy of ATP hydrolysis used during transcription. Would its hydrolysis lead to a further change of conformation of the TP? Or how do you envision the use of the ATPase to help in transcription?

Yes, we observed significant outwards movement (Figure 3) of TP (please see the subsection entitled “Structure changes in t-CPV and discovery of an ATP-binding site in TP”). We show that CPV capsid underwent two-step size expansion: one small (from unliganded CPV to S-CPV, therefore no ATP hydrolysis) and one large [from S-CPV (without hydrolysis) to t-CPV (with hydrolysis), and from S-CPV (without hydrolysis) to SGA-CPV (with hydrolysis)]. The energy released is used for these conformation changes.

*Currently, the data do not justify the title* “*ATP-binding mediates RNA transcription and capping in a dsRNA virus*”.

With the affirmative answers to the above two questions, we hope the reviewers are less concerned about our title. In the revised manuscript, we have changed the title to the following: “A putative ATPase mediates RNA transcription and capping in a dsRNA virus” to soften our tone.

More specifically, the following four facts support our conclusion that the sub-domain harboring the newly discovered ATP-binding site is likely an ATPase that mediates the activation of mRNA transcription. First, our results show that CPV’s ATPase activity and ATP binding to the putative ATPase sites are both SAM-dependent. Second, the two non-transcribing CPV (i.e., SGA-CPV and SG-CPV) have similar structures to that of transcribing CPV (t-CPV), indicating that the conformation changes observed in t-CPV are not a consequence, but rather a trigger of RNA transcription. Third, previous biochemical studies have shown that the mRNA transcription of CPV is SAM-dependent and is specifically coupled to ATP hydrolysis. Fourth, the removal of turret in orthoreovirus leads to loss of mRNA transcription activity (please see the subsection headed “Demonstration of viral ATPase activity and its SAM-dependence”). This cumulative evidence strongly suggests that the large sub-domain of CPV TP is likely an ATPase that plays a role in the activation of mRNA transcription.

*2) The description of the actual conformational changes is difficult to follow, and at points appears contradictory. The figures are also not very helpful in showing the conformational change. Morphings of the particle from one state to the others, provided as supplementary animations, would help a lot*.

We now provided two new movies (Movie 2 and Movie 4). These movies show morphing of CPV particle from CPV to S-CPV and from S-CPV to t-CPV.

3) Also, the observed conformational differences among some of these maps (e.g. S-CPV and CPV) are very small to support the hypothesis that 'the enlarged capsid may facilitate dsRNA template movement'. Could small differences in magnification have occurred among the different data sets and provide an alternative explanation for the enlarged capsid size? Could you also provide an estimate of the internal cavity in cubic Angstroms for each of the relevant reconstructions?

As indicated above, in the response to point 1, we report two-step conformational changes – one small and one large. The reviewer is correct that the conformational change from unliganded CPV to S-CPV is small. However, the “enlarged capsid” was not referring to this small change but rather to the large change (up to 9Å, see Figure 3) from S-CPV to t-CPV. The estimated internal volume of unliganded CPV and t-CPV is 73,900,000 Å^3^ and 77,500,000Å^3^, respectively. This is an ∼5% increase in volume.

Although the small change (i.e., that from CPV to S-CPV) is not relevant to our hypothesis, we trust the observed change is real and is unlikely due to magnification variation for the following two reasons. First, the slight expansion of viral capsid from unliganded CPV to S-CPV is anisotropic (The region proximal to the five-fold axis has the largest change of ∼1Å while the regions around the two- and three-fold axes have the smallest changes of ∼0.5Å.). Second, unlike the small changes between unliganded CPV and S-CPV, the structures of A-CPV and G-CPV are identical, so are the structures of t-CPV and SGA-CPV.

*4) It is not clear how the location of the magnesium location is identified. Whether the typical magnesium coordination geometry is satisfied, and how tentative is the assignment needs to be addressed*.

We identified the magnesium based on the density and chemical geometry. The modeled magnesium and GTP coordination has the cis bidentate conformation, which is the most commonly observed formation. We recognize that our maps are at about ∼3Å resolution and still not high enough to be very definitive for Mg modeling. To soften the tone, we have deleted Mg^2+^ from the Abstract and added “putative Mg^2+^ bound” in the revised manuscript (second paragraph of the subsection headed “The putative viral ATPase regulates the activities of GTase and MT-1”).

*5) In the t-CPV structure: it looks like the transcripts are getting extruded, but it appears there is no density that is observed in the turrets that can be assigned to exiting RNA molecules. Considering that endogenous transcription is a dynamic process, one would expect some of the structural features in the exit pathway should be smeared; however, there is no discussion of this in the manuscript*.

Good point. We have added the following statement (to the subsection “Structure changes in t-CPV and discovery of an ATP-binding site in TP”): “However, no mRNA densities are visible in our icosahedral reconstruction because these RNA molecules are transcripts of different genomic segments at different stages of the dynamic transcription process and are smeared by averaging.”

The density in the presented figures and movie looks quite good, so the following comments may not be that relevant to this particular paper. However, it is good practice to show clarity in all manuscripts about how potential overfitting was assessed in both cryo-EM map refinement and the atomic model refinement inside that map.

*6) You state that 'We have previously shown that our common-lines based programs do not suffer from the problems of over-fitting or model bias (*[47]*)'. However, the*
[47]
*paper doesn't explicitly mention overfitting, although it does show a gold-standard FSC curve in the supplementary information. Does that mean that you used gold-standard refinement procedures in the calculation of these maps? If so, you should mention this explicitly instead of citing the*
[47]
*paper. If you didn't, then refinements should be redone in the gold-standard manner (or by limiting the resolution used in refinement). Also, it would be good to show (gold-standard) FSC curves for each reconstruction in*
Figure 1—figure supplement 1.

*And*:

*7) R-factors are reported for the refined atomic models. Also in this refinement overfitting can be an issue. Without proper cross-validation, the R-factor curves in*
Figure 1—figure supplement 1
*cannot be trusted. You could either choose to limit the resolution used in the model refinement (and then assess the falloff of the R-factor or the FSC between the model and the map beyond that resolution, i.e. the predictive power of the model), or you could refine a scrambled version of the model in one of the gold-standard half-maps and then calculate FSC/R-factors between that model and that same half-map as well as FSC/R-factors between that model and the other half-map. The difference between these 2 curves is an indication of the amount of overfitting in the model. In the absence of overfitting, the 2 curves should overlap*.

We thank you for recognizing the quality of the maps. To provide further validation, we took advantage of the existence of identical structures of t-CPV and SGA-CPV (both independently determined) and calculated the FSC curves between the SGA-CPV map and the SGA-CPV model and that between the SGA-CPV map and the t-CPV model. Because these structures were independently determined, they are essentially the same as “gold-standard” FSC (30). These analyses further support our conclusion that our reconstructions do not have over-fitting (Figure 1—figure supplement 1 in the revised manuscript).